evolution, palaeontology, systems biology

birds, ontogeny, macroevolution, skull, shape, morphometrics

**Author for correspondence:**
Guillermo Navalón
e-mail: guillermo.navalon@earth.ox.ac.uk

# Craniofacial development illuminates the evolution of nightbirds (Strisores)

Guillermo Navalón[1,2,3], Sergio M. Nebreda[2], Jen A. Bright[4], Matteo Fabbri[5], Roger B. J. Benson[1], Bhart-Anjan Bhullar[5], Jesús Marugán-Lobón[2,6] and Emily J. Rayfield[3]

[1]Department of Earth Sciences, University of Oxford, Oxford, UK
[2]Unidad de Paleontología, Departamento de Biología, Universidad Autónoma de Madrid, Madrid, Spain
[3]School of Earth Sciences, University of Bristol, Life Sciences Building, Bristol, UK
[4]Department of Biological and Marine Sciences, University of Hull, Hull, UK
[5]Department of Earth and Planetary Sciences and Peabody Museum of Natural History, Yale University, New Haven, CT 06520, USA
[6]Dinosaur Institute, Natural History Museum of Los Angeles County, Los Angeles, CA, USA

 GN, 0000-0002-2447-1275; SMN, 0000-0001-8689-1972; JAB, 0000-0002-9284-9591; MF, 0000-0002-1257-1594; RBJB, 0000-0001-8244-6177; B-AB, 0000-0002-0838-8068; JM-L, 0000-0002-3766-8560; EJR, 0000-0002-2618-750X

Evolutionary variation in ontogeny played a central role in the origin of the avian skull. However, its influence in subsequent bird evolution is largely unexplored. We assess the links between ontogenetic and evolutionary variation of skull morphology in Strisores (nightbirds). Nightbirds span an exceptional range of ecologies, sizes, life-history traits and craniofacial morphologies constituting an ideal test for evo-devo hypotheses of avian craniofacial evolution. These morphologies include superficially 'juvenile-like' broad, flat skulls with short rostra and large orbits in swifts, nightjars and allied lineages, and the elongate, narrow rostra and globular skulls of hummingbirds. Here, we show that nightbird skulls undergo large ontogenetic shape changes that differ strongly from widespread avian patterns. While the superficially juvenile-like skull morphology of many adult nightbirds results from convergent evolution, rather than paedomorphosis, the divergent cranial morphology of hummingbirds originates from an evolutionary reversal to a more typical avian ontogenetic trajectory combined with accelerated ontogenetic shape change. Our findings underscore the evolutionary lability of cranial growth and development in birds, and the underappreciated role of this aspect of phenotypic variability in the macroevolutionary diversification of the amniote skull.

## 1. Introduction

Phenotypic macroevolution in vertebrates is generally studied through the lens of adult morphology. However, adult morphology ultimately emerges from variations in the developmental mechanisms unfolding through ontogeny. Development can impact morphological evolution in several ways [1]. One useful distinction among mechanisms of developmental evolution concerns whether those affected the relative timing of developmental events (i.e. heterochrony) or whether they affected other aspects of development (i.e. non-heterochronic mechanisms) [2]. The quantitative formalization of evolutionary comparisons of ontogenetic trajectories by Alberch *et al.* [3], and much subsequent work (reviewed by e.g. Zelditch *et al.* [4]), allows for the exploration and characterization of patterns of the evolution of development and underlying mechanisms using shape analysis. Specifically, the relationships between ontogenetic changes in shape and size, their relationship to individual age, and how those change along the branches of a phylogeny can be used to quantify ontogenetic trajectories and their evolutionary history ([4]; figure 1*a,b*). Therefore, the comparative study of ontogenetic

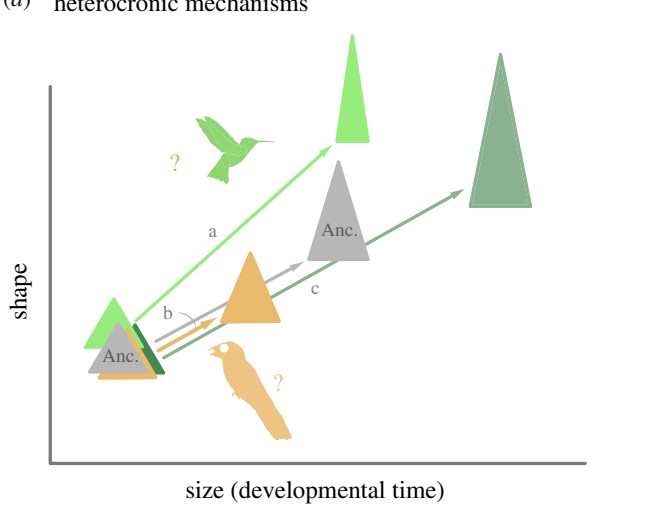

(a) heterocronic mechanisms

shape

size (developmental time)

a. acceleration (peramorphosis)
b. progenesis (paedomorphosis)
c. hypermorphosis (peramorphosis)

(b) non-heterocronic mechanisms

size (developmental time)

a. convergence (neomorphosis)
b. parallel trajectories (neomorphosis)
c. scaling†
d. divergence (neomorphosis)

(c) biological traits affecting hummingbird craniofacial evolution

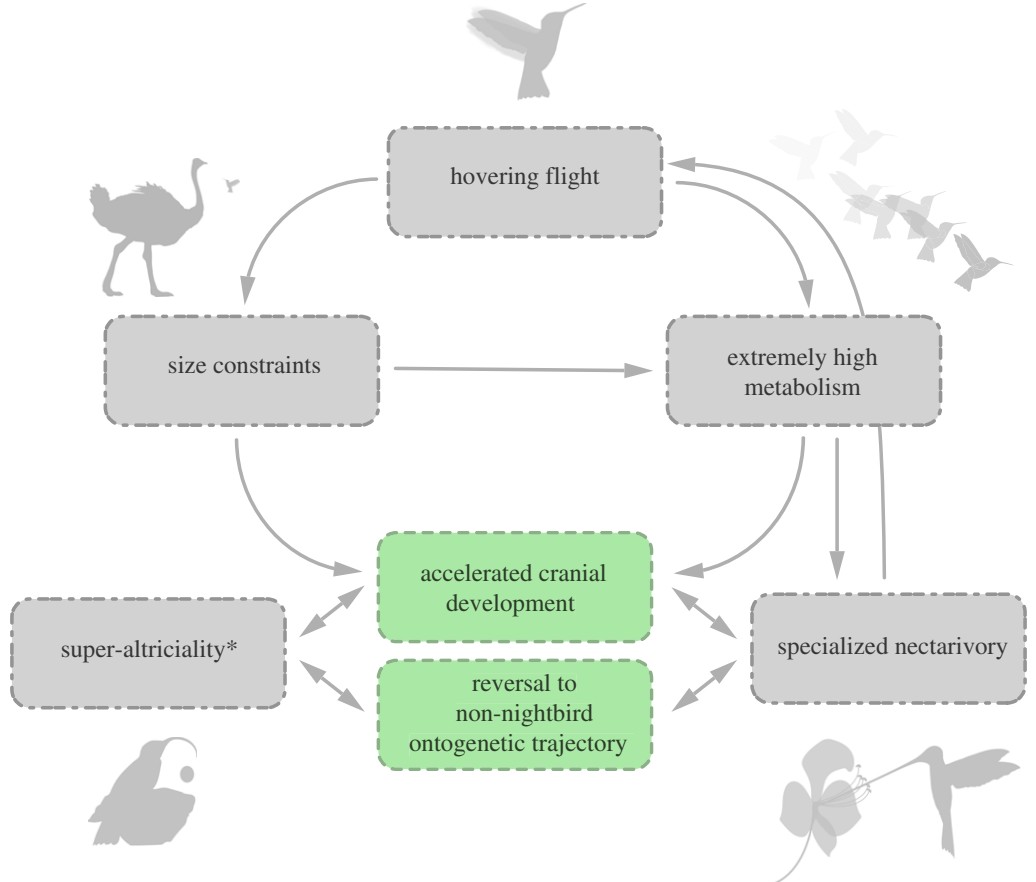

**Figure 1.** Expected morphometric relationships for selected mechanisms of evolution of developmental change and biological traits linked with craniofacial divergence in hummingbirds. Idealized representation of the effects in shape and size of selected mechanisms of evolutionary developmental change and including different kinds of heterochrony (a), scaling and neomorphosis (b). Changes in morphology and size are displayed with triangles for simplicity. Ancestral hypothetical development trajectory in grey, descendent development trajectories in different colours. Orange/ochre ontogenetic trajectory (underlined by the silhouette of a potoo) in part (a) shows the expectation of one kind of paedomorphism (progenesis) as hypothesized to have affected the broad skull nightbirds. Similarly, green ontogenetic trajectory (underlined by the silhouette of a hummingbird) portrays the ontogenetic trajectory hypothesized for by hummingbirds. †Scaling can produce isometric gigantism or miniaturization. Modified from [5]. (c) Diagram of the main factors hypothesized in this study to be connected to the evolution of the divergent cranial morphology in hummingbirds, and the suggested relationships between them. In green are alterations in mechanisms of craniofacial ontogeny inferred in this article to have occurred in the lineage leading to crown-hummingbirds (Trochillidae). In grey are ecological, physiological or life-history traits. The asterisk indicates that altriciality has been linked with both accelerated cranial development and increased overall lability of development. (Online version in colour.)

trajectories can yield great insight on the processes underlying patterns of phenotypic evolution.

The origin of the skeletal architecture, including the skull, of crown birds has been linked with the combined effects of both heterochronic and non-heterochronic mechanisms [6–8]. However, so far, there is only indirect evidence that growth-related developmental processes may have been important drivers of skull evolution during the subsequent diversification of the bird crown group [9–11].

We explore variation in ontogenetic trajectories of craniofacial morphology across nightbirds (Strisores), and their relationship with the evolution of adult morphology. Strisores, a well-established clade of crown birds [12,13] encapsulates an unusual diversity of skull morphologies, ecological and life-history traits [14]. Most nightbirds display a unique cranial morphology characterized by wide and flat braincases, beaks, and palates and enlarged orbits [15]. This morphology has been linked with cranial adaptations towards specialized foraging of small animals, in particular, aerial hawking (i.e. catching insects on the wing) in the nocturnal nightjars, owlet-nightjars and potoos and the diurnal swifts and tree swifts [16,17]. However, this cranial configuration also tends to co-occur with weak ossification in the braincase and it is morphologically reminiscent of immature birds, raising the question of whether heterochronic shifts in ontogenesis contributed to producing the apparent 'juvenilized' [2] cranial morphology in these lineages.

By contrast, hummingbirds (Trochillidae), phylogenetically nested within Strisores as the sister taxon of swifts, exhibit a very different cranial anatomy consisting of a globular braincase and elongated rostra regarded primarily as an adaptation to meet the functional demands of specialized nectarivory [18] and it is well-established that hummingbird beak shapes coevolved to some extent with floral morphologies [19,20]. Hummingbirds initially diverged from a swift-like cranial morphology [21,22] and they, therefore, provide a clear example of niche expansion (*sensu* [23]) followed by phenotypic diversification into a completely novel range of bill morphologies [24]. However, the factors linked with this initial divergence from a more typically nightbird cranial morphology are unknown. Interestingly, hummingbird hatchlings emerge from the egg with a superficially similar head shape to swifts and acquire their adult morphology in just a few weeks post-hatching [14], seemingly recapitulating the evolutionary trend observed in the fossil record [21] and suggesting an important role for developmental variations in this transition.

Furthermore, nightbirds display a wide range of adult sizes and metabolic requirements [14], and a considerable variety of life-history traits, with super-altriciality (hatchlings are comparatively underdeveloped) originating at least twice in the clade [25] and once coinciding with the origin of super-aerial locomotion in swifts and hummingbirds [17]. All these factors are likely to impinge upon the evolution of craniofacial development in the clade with possible effects in the patterns of macroevolution, particularly in the very divergent hummingbirds (figure 1*c*).

Here, we use three-dimensional shape analysis (geometric morphometrics) and phylogenetic comparative methods on a broad sample of extant and ancestral (inferred) ontogenetic craniofacial trajectories to ask: (i) is the origin of the typical-nightbird cranial morphology linked with heterochronic shifts connected with paedomorphosis? (ii) was the divergence of hummingbird cranial shape from that of other nightbirds facilitated by changes to ontogenetic trajectories? and (iii) does patterns of ontogenetic diversity among Strisores reflect the effects of other ecological or life-history traits?

# 2. Material and methods

## (a) Database, ontogenetic information

Our dataset includes 112 specimens belonging to 36 species, encompassing all the extant families of strisoreans plus five outgroups of non-neoavian birds assembled to provide information on the ancestral condition for Strisores: three galliforms and two palaeognathans (electronic supplementary material, dataset S1). Adult and immature specimens at several ontogenetic stages, including embryos, were obtained from museum collections or donated as natural casualties from farms or laboratory specimens. In total, partial ontogenies of 12 species were compiled (electronic supplementary material, dataset S1).

Because access to precise age information is rare for museum specimens, size is generally used as a standard proxy of developmental time in ontogenetic studies [4]. Additionally, the age at death for each of the farm natural casualty specimens used was documented (i.e. *Gallus* and *Struthio*, electronic supplementary material, dataset S1) or estimated using plumage information (see the electronic supplementary material, extended methods). This information was used to gain visual insights on the interplay between shape, size and developmental time.

We used laser surface scanners (most skeletal preparations) and several computed tomography (CT)-scanners (skins, fresh specimens and some skeletal preparations) to digitize cranial osteology as three-dimensional surfaces upon which shape analysis was subsequently conducted (details of scanning parameters and segmentation in the electronic supplementary material, extended methods). Surfaces for each specimen used in this study can be accessed as .ply files following this link: (http://data-bris.acrc.bris.ac.uk/deposits/21u51nyztyr2h2vseagk3o9jb5). Additional three-dimensional models were sourced from the project Phenome10 K (available online at http://phenome10k.org/).

## (b) Phylogenetic hypothesis and shape analysis

A time-calibrated maximum clade credibility (MCC) phylogeny of the 36 species included was generated using TREEANNOTATOR [26] from a population of 10 000 'Hackett's backbone stage 2 trees' (sourced from the in-built tools at www.birdtree.org; for the full details of the tree construction methods, see [27]). Branch lengths were set equal to 'Common ancestor' node heights (electronic supplementary material, figure S2; see extended methods for comparisons of our topology). Although the phylogenetic hypothesis adopted here for some strisorean lineages has been disputed by recent studies, alternative topologies would not change our conclusions (electronic supplementary material, extended results).

A set of seven landmarks and three curves, with five sliding semi-landmarks, was digitized in the best-preserved half of each skull (figure 2, electronic supplementary material, table S1) using LANDMARK EDITOR [28]. The landmarks of the palate and occiput regions (landmarks 8, 9 and 10, electronic supplementary material, figure S1a) were only digitized in the better preserved and generally more ossified adult specimens, to explore the phylogenetic patterns of cranial shape evolution in greater detail (figure 2*b*). Coordinates from the hemi-skulls were mirrored to obtain the bilaterally symmetrical full set of coordinates using FILE CONVERTER (available online at http://www.flywings.org.uk/fileConverter_page.htm), eliminating the asymmetric component of variation from downstream analyses. The shape

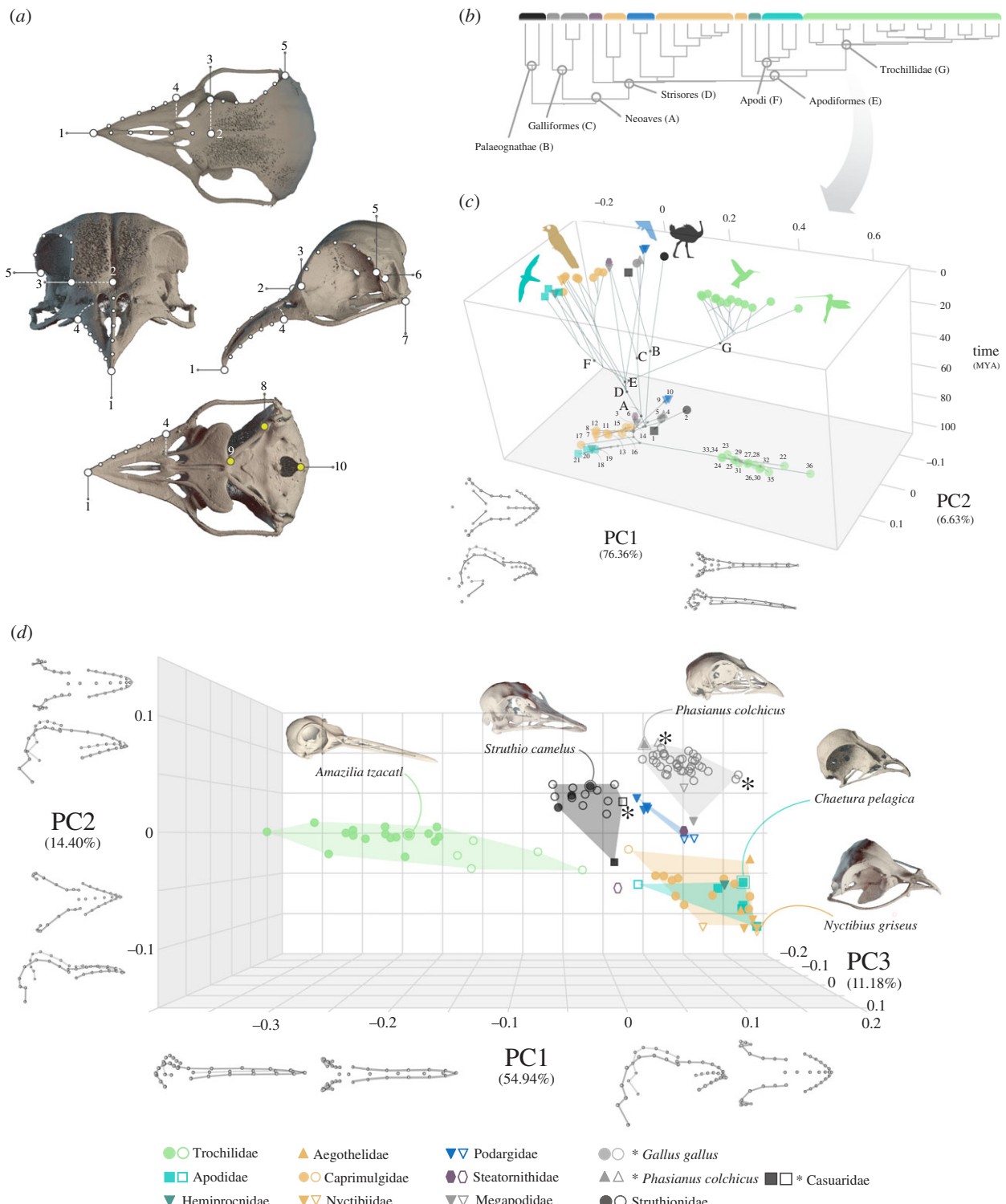

**Figure 2.** Patterns of morphological variation of the skull in Strisores and non-neoavian birds. (*a*) Landmarks and semi-landmarks used in this study placed over the digitally coloured skull of a large-tailed nightjar (*Caprimulgus macrurus*). Landmarks in yellow were only placed in adult specimens. (*b*) Simplified phylogenetic hypothesis for all the taxa included in this study (*n* = 36) (more detailed version in the electronic supplementary material, figure S1). (*c*) Chronophylomorphospace of time and the first two principal components (PCs) of skull shape (including all the landmarks in part *a*) of the species means of adult taxa included in this study, including maximum-likelihood ancestral shape estimates generated using plotGMPhyloMorphoSpace function from the R package geomorph v. 3.0.7. PC1-2 phylomorphospace shaded in the bottom of the three-dimensional plot with numbers corresponding to taxa in the electronic supplementary material, figure S1. Letters next to the nodes belong to clade names as in part (*a*). (*d*) PCA plot of the three main axes of cranial shape variation including all the specimens in our sample. Filled symbols represent adults, hollow symbols represent immature specimens and asterisks indicate late-stage perinatal embryos. Symbol legend next to clade names is followed across the study. (Online version in colour.)

was extracted thereafter using a generalized Procrustes analysis in geomorph v. 3.0.7. [29]. The minimum bending energy method was used to slide the semi-landmarks, as it is generally better suited to accommodate large variation in shape than other sliding criteria in the software used here (e.g. [30]).

Centroid size (CS) was used as a proxy of cranial size. Although CS is a widely used proxy of size in landmark-based geometric morphometrics, it provides skewed size estimates when configurations exhibit large local variation skewed in one spatial dimension (e.g. elongation; [31]). Our sample

encompasses both extreme brevirostral (*Nyctibius griseus*) and longirostral species (*Ensifera ensifera*), our CSs were, therefore, computed excluding the beak region landmarks (figure 2*a*; electronic supplementary material, table S1).

All downstream analyses were conducted using the R packages geomorph v. 3.0.7 and v. 3.3.1 and RRPP [32] except indicated otherwise. Significance of differences was always tested using 1000 permutations of the relevant variables.

We conducted three principal components analyses (PCA) using different sets of specimens and shape coordinates using the gm.prcomp function: (i) a PCA using the whole landmark configuration (figure 2*a*) and species means for adult specimens used to visually explore evolutionary variation in adult cranial morphology among Strisores (figure 2*b*); (ii) a second PCA including all specimens in our sample but excluding landmarks 8, 9 and 10 which could not be placed in juveniles (figure 2*a*) to visually explore the interplay between ontogenetic and evolutionary cranial shape variation; and (iii) a third PCA including only landmarks from the braincase region to exclude the possibility that observed patterns are created by local variation in the beak region (electronic supplementary material, figures S3 and S4). Because PCA plots can only display the tangent shape space contained within three axes of shape variation, we also conducted an unweighted pair-group average (UPGMA) cluster analyses using Procrustes coordinates (shape) for the two latter subsets (all the specimens, whole skull configuration and only braincase landmarks) to explore the shape variation in the total shape space using PAST v. 3.15 [33]. UPGMA cluster analysis outputs a branching diagram that summarizes the Procrustes linear distances among specimens.

## (c) Comparative analyses of ontogenetic trajectories

We used a combination of the analysis of allometry and phenotypic trajectories to test our evo-devo hypotheses, following several recent studies (e.g. [34–36]) which make explicit predictions about the expected relationships of shape and size under the main scenarios of evolutionary changes in development (figure 1*a,b*). Analytically we used a Procrustes ANOVA distance-based approach [37].

To quantify patterns of ontogenetic allometry in the 12 species for which ontogenies were obtained we used ordinary least squares (OLS) regression of skull shape as a function of skull CS and species identity using procD.lm and pairwise functions [34] and tested for pairwise differences. Specifically, we tested for differences in two allometric descriptors: (i) slope vector length, the amount of shape change per unit of size change; and (ii) slope vector angle, how shape changes in relation to size during growth. We could not use phylogenetic comparative approaches for this analysis, because immature and adult specimens of the same species cannot be coerced into a single phylogeny. We also conducted an OLS regression of shape as a function of CS and clade using all the specimens in our dataset.

To test for differences in the direction and total amount of ontogenetic shape change among species we used phenotypic trajectory analysis (PTA) [36]. PTA was implemented using the function trajectory.analysis which computes: (i) the amount of shape change; and (ii) direction for each ontogeny, and statistically tests for pairwise differences. PTA requires equivalent numbers of ontogenetic stages to be compared; because the 12 ontogenies are unevenly sampled (some encompassing 10–30 individuals (e.g. *Struthio camelus*) and some encompassing only two specimens (e.g. *Steatornis caripensis*), only the adults and the earliest-stage per species were used. Ontogenetic trajectories for PTA were, therefore, linear trajectories uniting two shapes. Although the earliest stages used were developmentally similar in the majority of the ontogenies compared, there are some instances in which this is not the case (electronic supplementary material, dataset S1).

We, therefore, focus our interpretations on ontogenetic trajectories for which the youngest specimens were estimated as being in the first week after hatching, or right before hatching (from plumage traits in skinned specimens, see the electronic supplementary material, figure S2 and datasets S1, S3) or known (for farm and laboratory specimens). When several earliest-stage or adult-stage individuals were available, the mean shapes were used. Additionally, to test for differences in ontogenetic disparity among clades, we computed Procrustes variances for various taxonomic groups of earliest-stage individuals and compared them to adults of the same groups, testing the significance of the observed differences using the function morphol.disparity.

Because the effects of mechanisms of developmental evolution make explicit predictions about the relationship among descendant and ancestral ontogenies (figure 1*a,b*) we reconstructed ancestral cranial ontogenies for clades of interest (Aves, Strisores and Apodiformes) using maximum-likelihood implemented within the function gm.prcomp. We compared inferred ancestral ontogenies with descendant ontogenies using PTA and ontogenetic allometry following the same methods as above. Ancestral hatchling morphologies and cranial sizes were estimates using only specimens of young hatchlings (defined above). Ancestral adult morphologies and cranial sizes were reconstructed using these same taxa and, also, using all the adults in the database.

Finally, to ask whether the divergence of hummingbird cranial shape from other nightbirds was facilitated by changes to ontogenetic trajectories, and whether evolutionary changes in cranial shape in hummingbirds are recapitulated over hummingbird ontogeny or not, we compared inferred ontogenetic changes with evolutionary change along phylogenetic branches, focusing on evolutionary change at the root node of hummingbirds (Trochillidae) basal node.

To gain further visual insight on the interplay of actual age with the ontogenetic relationship between skull shape and CS, we incorporated inferred ages into several of the plots resulting from the analyses, and plotted regression scores of shape from the OLS regression including all individuals, with CS values and age.

## 3. Results

The first two principal components (PC1 and PC2) of adult skull shape variation account for approximately 83% of total shape variation (figure 2*b*) and groups occupy broadly distinct areas in this morphospace. Skull evolution in hummingbirds (Trochillidae) is characterized by an abrupt and large early divergence towards positive values of PC1, characterized by a basal shift towards a longirostrine morphology accompanied by further changes in palatal and neurocranial morphology. PC2 mainly describes differences in the relative size and orientation of the orbits, the relative size and shape of beak, and palatal morphology and orientation; broadly separating owlet-nightjars (Aegothelidae), and swifts and tree swifts (Apodi) with broader beaks and braincases and expanded orbits, from palaeognathans and galliforms, with narrower and deeper beaks and braincases, and relatively smaller orbits. Frogmouths (Podargidae) and the oilbird (Steatornithidae) seem to slightly diverge in shape from the rest of strisoreans approaching the cranial morphologies of galliforms and palaeognathans. Including immature specimens produces similar patterns of shape variation, although less of the total shape variation is condensed in PCs 1–2 (figure 2*c*). Immature hummingbirds have intermediate shapes between adult hummingbirds and other birds. Local variation in braincase shape produces largely comparable patterns among groups to whole skull shape

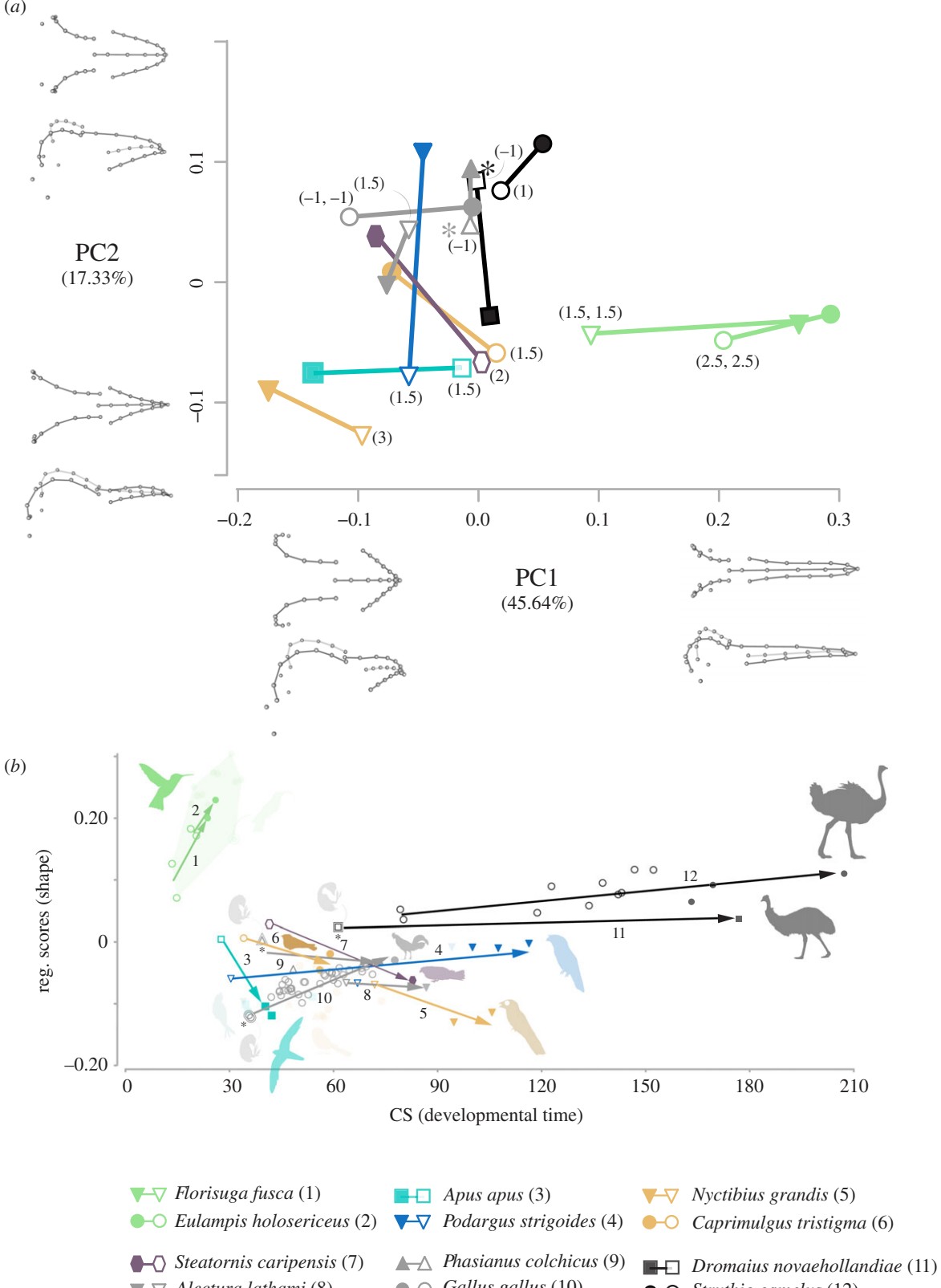

**Figure 3.** Comparisons of ontogenetic shape trajectories and ontogenetic allometric trajectories among the 12 species included in this study. (*a*) Principal components analysis plot of the two main axes of cranial shape variation encompassing the 12 ontogenies included in this study. Individual dots are mean values of the different specimens from the youngest stages and adult stages, numbers in brackets next to the symbols represent the precise or approximated age in weeks (see the electronic supplementary material, figure S5 for comparisons with ancestral ontogenies and evolutionary change along the two branches selected in this study, see methods). (*b*) Multivariate regression of shape on CS for the whole dataset with the 12 ontogenies included in this study highlighted (see the electronic supplementary material, figure S6 for the regression with the whole dataset). (Online version in colour.)

variation (electronic supplementary material, figure S4). Cluster analysis dendrograms show that the total phenotypic distances between all specimens largely corroborate these patterns (electronic supplementary material, figure S3).

The PCA plot of ontogenetic trajectories visually reveals all the earliest immature strisorean individuals share a common region of the morphospace, initially similar in shape among them and distinct from the non-strisorean

taxa, from which they widely diverge through ontogeny, with hummingbird ontogenies progressing through shape space in a highly divergent direction from other strisoreans (figure 3a; electronic supplementary material, table S2). The amount of total shape variation during ontogeny ranges from 0.24 Procrustes shape distance in the tawny frogmouth (*Podargus strigoides*) to only 0.08 in the common ostrich (*Struthio camelus*) which is the only ontogeny that exhibits significantly lower shape changes and also includes a near hatchling (electronic supplementary material, table S3). By contrast, most of the superficially 'juvenilized' strisorean taxa display substantial craniofacial shape change during ontogeny. The common swift (*Apus apus*) and the black jacobin (*Florisuga fusca*) exhibit among the largest shape changes during ontogeny. More detailed comparisons of angles between ontogenetic shape trajectories suggest the black jacobin (the hummingbird species of the two included here that includes the earliest immatures) and the chicken (*Gallus gallus*), exhibit similar ontogenetic shape trajectories and significant or near-significant differences in ontogenetic trajectory from most of the remaining strisoreans (electronic supplementary material, table S3). The remaining taxa do not exhibit significant differences in the pattern of craniofacial ontogenesis, except for the emu (*Dromaius novaehollandiae*) (electronic supplementary material, table S3).

Both hummingbird ontogenies exhibit a higher ratio of shape change per unit of size change (i.e. slope vector lengths) than all the remaining species. The common swift displays the second highest ratio of allometric change, significantly different from nearly all the other species. By contrast, the ostrich displays the lowest allometric ratio, indicating near isometric growth. However, pairwise comparisons between allometric vectors (using all the specimens for each of the 12 taxa) (figure 3b) reveal that the allometric patterns (i.e. slope vector angles) are not statistically different among most of the species (electronic supplementary material, table S6) except from the emu, whose adult morphology is broadly similar in shape to most strisorean immatures. OLS allometric regression using all specimens in the dataset are congruent with these ontogenetic observations (electronic supplementary material, figure S6 and table S5).

We find evidence that the ancestral ontogenetic trajectory for Strisores is more similar in pattern to descendent non-hummingbird strisoreans than it is to hummingbirds and non-strisoreans (electronic supplementary material, tables S4 and S5). However, we did not find support for ontogenetic truncation in any of the descendent strisorean species as predicted by progenesis, a specific type of paedomorphosis (figure 1a). Furthermore, although the ancestral ontogenetic pattern of Apodiformes is not significantly different to those of any of the descendant species, the three descendant species exhibit significantly extended ontogenetic shape change (electronic supplementary material, table S5) and significantly higher allometric slopes (electronic supplementary material, table S7), supporting our observation of accelerated shape change during late ontogeny (figures 1a and 3b).

Finally, evolutionary change between the ancestral shapes for the node of Apodiformes (crown swifts and tree swifts plus hummingbirds) and crown-hummingbirds is very similar to the ontogenetic shape change in both hummingbird species (angles below approx. 41°, electronic supplementary material, table S4) and it is significantly or near-significantly different from all the other strisorean ontogenies (including

the ancestrally inferred ontogeny for Strisores) except from the frogmouth (electronic supplementary material, table S5). Also, the pattern of evolutionary allometry in this branch of the phylogeny is significantly different to descendent ontogenies of both hummingbird species, chicken and ostrich suggesting that although the shape change is similar, the size change is reversed, underlining the size reduction pattern undergone along the stem lineage of hummingbirds.

## 4. Discussion

The high cranial disparity attained by nightbirds over their evolutionary history is mirrored by a significant diversity of ontogenetic trajectories, suggesting central roles for developmental processes in shaping macroevolutionary patterns in this clade of birds.

Anatomical similarities between the juveniles of many bird species and the adults of many nightbird species include the presence of a wide, short mouths and beak. We demonstrate that these result from convergent evolution rather than as products of a paedomorphic evolutionary processes. Indeed, ontogenetic changes to cranial shape during posthatching growth in swifts, nightjars, potoos and oilbirds (and frogmouths to some extent, see the electronic supplementary material, extended results) are surprisingly large, and divergent from the ancestral ontogenetic trajectory of birds (represented here by galliforms and palaeognathans and ancestral ontogenies), which involves anterior projection and lengthening of the beak region (figure 4) [25,38]. The ontogenetic trajectory seen in most groups of nightbirds is instead characterized by a progressive broadening of the skull and braincase. This probably originated at the base of the clade and can be regarded as an autapomorphy (figure 4; electronic supplementary material, figure S10). This developmental trajectory originated perhaps as a result of adaptation to the very specialized foraging ecologies of the adult individuals, which have been suggested to require wide flat beaks to facilitate large gapes for efficient insect collection, and expanded palatines to withstand the impacts of aerial insect prey [14,16,39]. The retention of a similar ontogenetic pattern in the frugivorous oilbird is probably owing to shared phylogenetic history as aerial hawking has recently been proposed as an ancestral trait for all nightbirds [40].

Hummingbirds display a much more pronounced ontogenetic allometry than any other lineage of birds studied here. However, ontogenetic shape change in hummingbirds is large and more similar to non-nightbird taxa. Furthermore, the ontogenetic trajectory of the hummingbird skull recapitulates the morphological changes experienced during the initial divergence of stem-hummingbirds from a more typical nightbird morphology [21,24]. Therefore, reversal to a more ancestrallike (i.e. non-nightbird) ontogenetic pattern together with a general acceleration of postnatal craniofacial growth was central to the initial morphological divergence of stem-hummingbirds from other nightbirds (figures 1 and 4). The rapidity of ontogenetic shape changes in hummingbirds is especially striking, occurring in less than two weeks of postnatal growth (electronic supplementary material, figures S8 and S9).

We also found evidence, although to a lesser degree, for the acceleration of craniofacial ontogeny in the common swift, the other apodiform species studied here. Craniofacial ontogenetic acceleration in these two lineages could be related to their

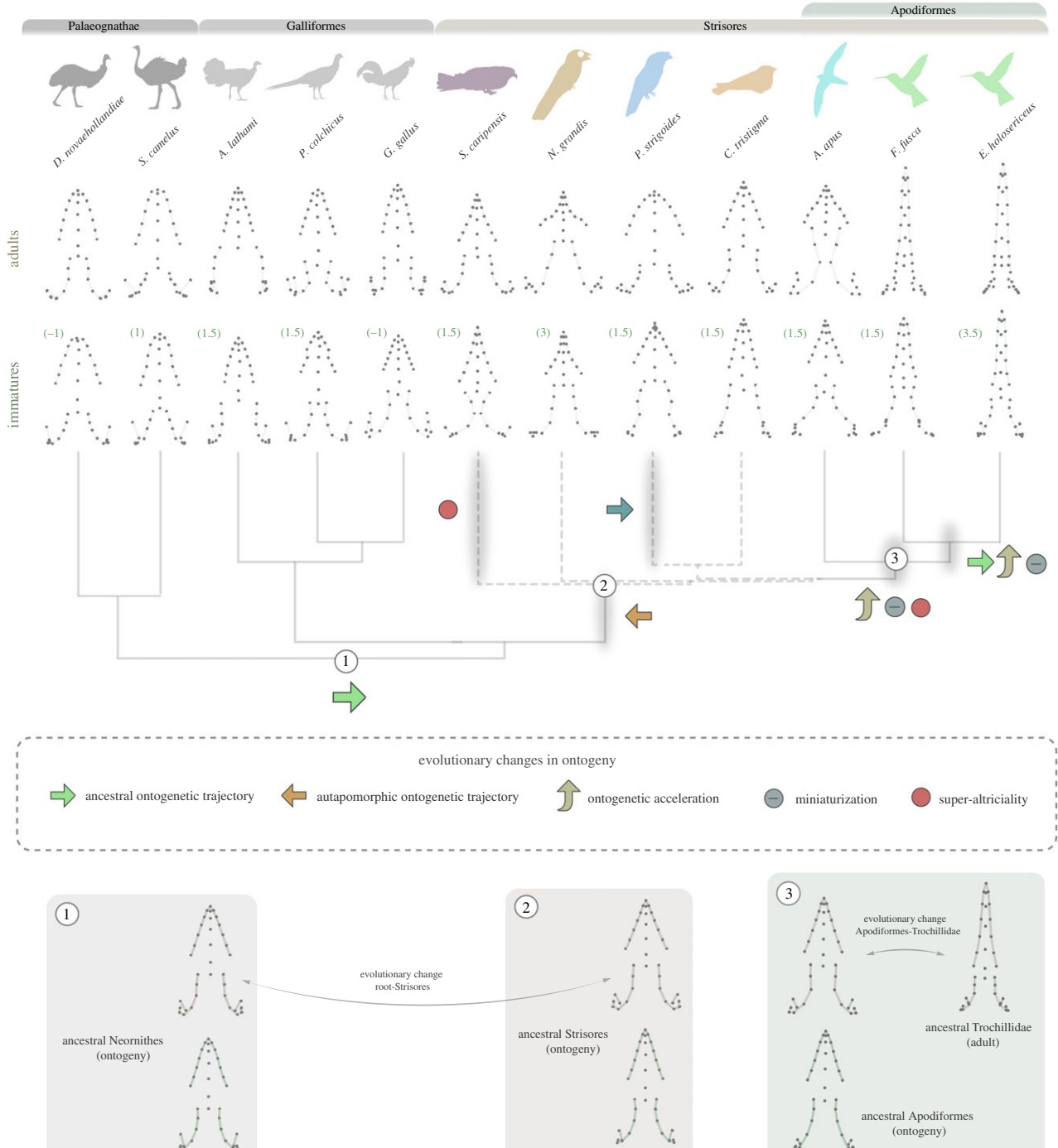

**Figure 4.** Cranial shape changes (dorsal view) along ontogeny of the 12 species included in this study, identifying the key evolutionary changes in ontogeny inferred during strisorean evolution, and reconstructed ancestral ontogenetic shape changes for three nodes. Depicted immatures are the earliest stages per species in our dataset. Estimated or known week-range age (see methods) is written in brackets next to the cranial shape of each immature specimen. Proposed evolutionary changes in craniofacial ontogeny occur along the shaded branches. Blue arrow in the branch leading to the Podargidae reflects that the reversal to the ancestral ontogeny is only partial. Ancestral reconstructed ontogenies for three key nodes: 1, Neornithes, 2, Strisores and 3, Apodiformes. Dashed lines in phylogeny reflect the current phylogenetic uncertainty of the branching pattern of several lineages of Strisores. Tree topology from the MCC tree. Lateral-oblique views in the electronic supplementary material, figure S7. (Online version in colour.)

super-altricial condition, an observation in agreement with previous studies showing accelerated development and greater developmental plasticity during postnatal growth in altricial birds (although the oilbird might be an exception, see [25]). An alternative, but not mutually exclusive, view is that maximum adult sizes might be bounded in swifts and hummingbirds by their very specialized hyper-aerial flight styles, which require high flight proficiency [14]. These constraints may have affected hummingbirds more intensely as their specialized hovering flight requires even smaller sizes to

be efficient (large hummingbirds operate close to the theoretical aerodynamic size limit for their flight style, [41]). Therefore, adult cranial morphology needs to be attained with minimal size change, imposing constraints on craniofacial development. Size reduction may, therefore, have contributed in some way to the ontogenetic acceleration seen in apodiforms, and particularly in the lineage leading to hummingbirds (figure 4; electronic supplementary material, figure S9, and see S10 for a detailed account of the macroevolutionary context incorporating stem lineages). In turn, the extremely small sizes of

hummingbirds might be allometrically related to the evolution of an extremely high metabolism [42], the highest for non-insect animals [14], which may had needed a nutrient-rich diet such as nectar to be energetically sustainable. As hummingbirds increased their reliance on nectar feeding over evolution, selection pressures may had favoured the synchronic development of the postcranial adaptations connected with the unique hovering flight style together with the elongation of the beak for accessing the nectar in the delicate angiosperm flower. However, elongation of the beak is incompatible with the miniaturization displayed by hummingbirds, as an upper bound to maximum adult size also affects the maximum size of the hatchling (the egg needs to be hatched by a small adult individual) and a longirostrine beak would not easily fit in a small egg (figure 1c). To circumvent this, we propose here that hummingbirds further accelerated post-hatching craniofacial development allowing, in turn, more longirostral adult forms to evolve within a scenario affected by this plethora of simultaneous trade-offs (i.e. constrained size, extremely high metabolism, specialized nectarivory, hovering flight; summarized in figure 1c).

Although all these factors might be linked during hummingbird evolution, the fossil record provides the ultimate test to ascertain the relative timing of the appearance of these traits along the stem of hummingbirds. Unfortunately, the most complete stem-hummingbirds, *Eurotrochilus* from the Oligocene of Europe, already exhibit a modern cranial and postcranial anatomy including many of the morphological adaptations linked to the specialized ecology found in modern representatives of the clade [43]. The only other stem-hummingbird known from complete specimens is the Early Eocene European genus *Parargornis* [15,21,22] which shares several postcranial traits shared with hummingbirds but exhibits very different limb proportions and plumage, and a remarkable swift-like cranial morphology, with a broad beak and skull, which suggests its ecology may have been very different to modern hummingbirds. Although the seemingly sudden evolutionary appearance of the suite of anatomical traits that characterize modern hummingbirds is probably a reflection of a patchy fossil record, if developmental changes are connected to the early evolution of the hummingbird skull, this morphological divergence may have also been rapid.

Our results show a previously unreported disparity of craniofacial ontogenetic patterns in birds (see more details in the electronic supplementary material, extended results), suggesting that evolutionary changes in development facilitated deep divergences in avian cranial morphology. However, we note the possibility that nightbirds show exceptional ontogenetic variability among birds. Our findings demonstrate the importance of the study of ontogenetic change beyond model organisms using current quantitative methods. Our study, and future studies of phylogenetically broader scope, provide a promising avenue to advance knowledge of the evolution of ontogeny and its contributions in defining large scale macroevolutionary patterns. Furthermore, our study highlights the importance of rare museum specimens for research, as an invaluable means (and sometimes the only way) to access elusive information from poorly known or endangered species.

Data accessibility. All data relevant to the reviewing process is provided here as electronic supplementary files (electronic supplementary material, datasets S1 and S2 and supplementary information). Additionally, the final clean three-dimensional meshes for all specimens can be downloaded from the Bristol University servers at: https://data.bris.ac.uk/data/dataset/2lu51nyztyr2h2vseagk3o9jb5.

Authors' contributions. G.N. established the original working hypotheses. G.N., J.A.B., J.M.-L. and E.J.R. designed the research. M.F., R.B.J.B. and B.-A.B. contributed with important specimens. G.N., E.J.R. and J.A.B. selected the remaining specimens in museum collections. G.N., S.M.N. and M.F. curated the specimens and obtained the data. G.N. did the analyses. G.N. wrote the original draft of the manuscript. G.N., S.M.N., J.A.B., M.F., R.B.J.B., B.A.B., J.M.-L. and E.J.R. contributed to writing the manuscript.

Competing interests. We declare we have no competing interests.

Funding. G.N. was funded by a grant from the Alumni Foundation of The University of Bristol and is currently funded by the European Union's Horizon 2020 research and innovation program 2014–2018 under grant agreement 677774 (European Research Council [ERC] Starting Grant: TEMPO). S.M.N. was funded by SRUK in agreement with Erasmus+ Placement Programme and an undergraduate stipend from the University of Bristol (ref JW/MM/JB/1870253) and is currently funded by a FPI-UAM 2019 doctoral scholarship from Universidad Autónoma de Madrid. E.J.R. and J.A.B. were funded by a grant from the Biotechnology and Biological Sciences Research Council (BBSRC, BB/I011668/1).

Acknowledgements. We thank Joanne Cooper, Judith White and Mark Adams (National History Museum at Tring, UK) for access to specimens. We thank Gavin H. Thomas (University of Sheffield, UK) for access to surface laser scanners and enlightening insights on both macroevolution and methodological choices. We are grateful to Tom Davies (University of Bristol, UK) for assistance during challenging micro-CT scanning and to Jennifer J. Hill (Smithsonian National Museum of Natural History, USA) for kindly micro-CT scanning additional specimens for us. We are very grateful to two anonymous reviewers whose comments made the article better. We thank Fernando Blanco, Iris Menéndez and Francisco J. Serrano for discussion on macroevolutionary implications and article narrative. We are also grateful to Lucía Balsa Pascual and Óscar Sanisidro for design advice that helped improve the quality of the graphic support.

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
