## [Peer Review File · Proceedings of the Royal Society B: Biological Sciences]

Review History

RSPB-2021-0181.R0 (Original submission)

Review form: Reviewer 1

Recommendation

Major revision is needed (please make suggestions in comments)

Scientific importance: Is the manuscript an original and important contribution to its field?

Excellent

General interest: Is the paper of sufficient general interest?

Excellent

Quality of the paper: Is the overall quality of the paper suitable?

Good

Is the length of the paper justified?

Yes

Should the paper be seen by a specialist statistical reviewer?

No

Do you have any concerns about statistical analyses in this paper? If so, please specify them explicitly in your report.

No

It is a condition of publication that authors make their supporting data, code and materials available - either as supplementary material or hosted in an external repository. Please rate, if applicable, the supporting data on the following criteria.

Is it accessible?

Yes

Is it clear?

Yes

Is it adequate?

Yes

Do you have any ethical concerns with this paper?

No

Comments to the Author

I have read the paper written by Navalon and collaborators entitled: "Craniofacial development illuminates the evolution of nightbirds (Strisores)". The aim of the paper is to elucidate whether heterochronic shifts during ontogeny contribute to producing an apparently juvenilized cranial morphology in birds. More specifically, they aim to assess whether nightbird cranial morphology is related to ontogenetic changes and whether these can also explain the cranial shape differences in hummingbirds. To do so, the authors used a 3D geometric morphometric approach and comparative methods on a unique data set composed of 112 specimens belonging to 36 species as well as a subset with several ontogenetic series for 12 species.

I think at this point it is really important to highlight how difficult it is to get access to ontogenetic series and how important this kind of studies are. This study has a high potential and would be a really nice contribution to Proceedings B. However, I think the manuscript needs a little bit of work as some parts remain unclear or can be simplified. I think the introduction is the weakest part of the paper and needs to be rewritten in order to better introduce the concept and more clearly introduce the aim of the paper. Currently, the introduction reads as a set of paragraphs not really linked to each other and is sometimes repetitive. All the elements are there, but a little bit of work on it would be nice in order to better introduce the concept (nicely and beautifully synthesized on Figure S1 and S11) and also to show how different factors can impact the cranial evolution of the incredible Strisores clade. The material and methods and results can be clarified by the addition of headers synthesizing what the authors want to test using which methods. I will give more information on the changes that I would like to see in the manuscript below, but overall, I would like to see this manuscript published in Proceedings B and I mainly recommend writing and clarification revisions.

The manuscript is not in the format for Proc B, so this needs to be changed.

Introduction:

I would like to see an introduction written around figure S1 and maybe figure S11 with some modifications (and thus the figures move into the main manuscript) in order to summarize the different factors which could shape the cranial evolution. The combination of both figures is a really nice illustration of the questions addressed in this paper on whether altriciality is related to both accelerated cranial development and increased overall lability of development.

I would start the introduction with the explanation of the main concepts and on how development can impact morphological evolution. Move directly onto the questions addressed in this study and the presentation of the amazing model group. In the current introduction, the predictions concerning the impact of the different life history traits (highly altricial, altricial) as

well as the other factors (such as specialized functional demands of nectarivory with high metabolism and so one) are not clearly made and are really missing. Yet, this is needed to understand the rest of the manuscript. Furthermore, because of the lack of a connection between each paragraph, I have the feeling that a new question is addressed into each paragraph. Then I would like to see the state of the art concerning this question. Here the authors can highlight that nearly no studies have looked at ontogenetic series and this is why their study is unique. I would also state why nearly no studies are looking at this. It is easy to say that other studies look at adults only, but it is also because ontogenetic series are not abundant in collections (which makes this kind of study complicated to realize).

I would end the introduction with a paragraph on what they will more specifically investigate and the predictions of the results following the literature. These predictions will help to structure the method section: for example: species with strong developmental constraints will show a lower disparity than those that are less constraint, and so one).

Line 47: references are needed

Line 43: the way that the word macroevolution is used here doesn't make any sense It should be use as 'pattern of evolution at the macroevolutionary scale'

Line 69: "hummingbird" need to be replace by hummingbird

MATERIAL AND METHODS:

I would mainly add headers summarizing what the authors want to test using which methods. For each section I would start by the aim of the analyses prior to giving the detail on the analyses. This will make the reading of the methods easier and will make it more straightforward to understand why which kind of method is performed and why.

I would move the phylogeny section into the methods section (it is currently in material).

One part needs to be clarified concerning the test of convergence / divergence. I may have missed something while reading, but it is unclear to me how the authors can test for shape convergence/ divergence while using disparity tests. For me, they can only assess difference in morphological diversity using this kind of test (see below for more details).

Line 117: change "followed" by "used"

Paragraph line 119-122: need to be rephrased in order to better link both sentences. Moreover, 'skewed' is used twice in the same sentence

Line 127: the script provided in supplementary information is not up to date in comparison to what they write in the main manuscript (the authors used the "plotTangentSpace" function in the script, which has been recently deprecated, and is now replaced by the "gm.prcomp" function). As the geomorph R package changes the name of their functions really often and the authors only used published functions in this manuscript, I don't see the point to provide the script. It may be confusing for the reader if they want to reproduce the analyses in few months, the script may be quickly unusable by anybody due to the changes of the names of the functions. Furthermore, nobody is asking to see screenshot of the button to use while they are using "PAST".

Line 164: the authors used a test of disparity in order to assess ontogenetic convergence/ divergence pattern among clades. Maybe I misunderstood something, but by doing that, I have the feeling that they can only assess whether the disparity is higher or lower in one clade in comparison to another (so just difference in shape variance and not difference in shape). To test that I would either do a test comparing the slope, or do a test of convergence (Stayton test or RRphylo test of convergence).

Line 167: function plotGMPhyloMorphoSpace is now Deprecated and defunct functions in geomorph

Line 172 "focussing" needs to be changed by "focusing"

RESULTS

Adding headers as in method section can be helpful.

DISCUSSION

Line 282 and 285: there is no fig. 4 in the manuscript.

Line 301: extra space between "(1-3)" and "."

Figure 1: the use of the letters (a., b., c., d.....) in both phylogeny+phyomorphospace (node labeling) and in the overall figure is confusing. It is possible to change them in the phylogeny+phyomorphospace by capitalized letters?

Supplemental information:

Table S3, the table caption is before the table whereas it is placed after for all the other tables.

Review form: Reviewer 2

Recommendation

Accept with minor revision (please list in comments)

Scientific importance: Is the manuscript an original and important contribution to its field?

Excellent

General interest: Is the paper of sufficient general interest?

Excellent

Quality of the paper: Is the overall quality of the paper suitable?

Good

Is the length of the paper justified?

Yes

Should the paper be seen by a specialist statistical reviewer?

Yes

Do you have any concerns about statistical analyses in this paper? If so, please specify them explicitly in your report.

No

It is a condition of publication that authors make their supporting data, code and materials available - either as supplementary material or hosted in an external repository. Please rate, if applicable, the supporting data on the following criteria.

Is it accessible?

Yes

Is it clear?

Yes

Is it adequate?

Yes

Do you have any ethical concerns with this paper?

No

Comments to the Author

This is an extremely interesting study that will contribute greatly to our understanding of the evolution and development of an unusual clade of birds, with implications for the macroevolution of vertebrates in general. The figures are well rendered and the interpretations seem to be reasonable given the findings presented. Accordingly, I have only a few minor concerns.

Firstly, I appreciate the repeated acknowledgements that the internal relationships within Strisores are controversial. However, I am not aware of any recent study that has recovered the topology used in this study, whereas White and Braun (2019) and Kuhl et al. (2021) recovered topologies congruent with those of Prum et al./Chen et al., and the preferred topology of Braun and Kimball (2021) differs from those only in the position of *Steatornis*. Given that these changes in tree topology are unlikely to change the study's conclusions (as stated in the supplementary material), I would not insist on rerunning the analyses under alternative phylogenetic hypotheses, but I recommend that this is given greater emphasis in the manuscript, preferably with at least a brief mention in the main text, e.g., "Although the phylogenetic hypothesis adopted here has been disputed by recent studies, alternative topologies would not change our conclusions (Supplementary Information)."

Although the discussion understandably focuses on hummingbird origins, the findings on frogmouths are also interesting, and I think further discussion of these results (at least in supplementary material) would make a good addition to the manuscript. In particular, it is mentioned several times that frogmouths exhibit only a "partial" reversal to the ancestral ontogenetic pattern. I would be interested in seeing further elaboration on what this entails (i.e. in what ways is frogmouth ontogeny similar to/different from those of other strisoreans/non-strisorean birds?). Also worthy of discussion might be potential implications of this finding for previous speculations that frogmouths evolved from aerial hawkers (Serventy, 1936; Chen et al.) and the timing of this transition based on fossils of stem-frogmouths (*Masillapodargus* and possibly *Fluvioviridavis*).

I find it odd that caprimulgids are color-coded as semi-altricial in Fig. S10, but are (to my knowledge correctly) listed as semi-precocial in Supplementary Dataset 2B. It is also not immediately clear to me what the arrows on the leftmost side of the figure indicate. Perhaps an explanation in the caption is warranted here.

Lastly, I offer a list of grammatical, stylistic, or typographic suggestions:

Line 65: The comma after "configuration" is not needed.

Line 74: I suggest "developmental variations".

Line 241: The comma after "extent" is not needed.

Line 266: Looks like a closing parenthesis is needed after citation (40).

Line 273: Should read "leading to hummingbirds".

Line 275: Should read "may **have**".

Fig. S4a: I suggest changing "most youngest specimens" to "the majority of the youngest specimens" (as in S4b), as the latter sounds less confusing.

Fig. S11: I recommend "In green **are** alterations [...]" and "In grey **are** ecological [...]", as this seems to read more clearly.

Supplementary Dataset 2B: Plumage details for *Eulampis holosericeus* should read "Pinkish **chick**", not "Pinkish chicken".

References

- Braun, E.L. and R.T. Kimball. 2021. Data types and the phylogeny of Neoaves. *Birds* 2: 1–22. doi: 10.3390/birds2010001
- Kuhl, H., C. Frankl-Vilches, A. Bakker, G. Mayr, G. Nikolaus, S.T. Boerno, S. Klages, B. Timmermann, and M. Gahr. 2021. An unbiased molecular approach using 3'UTRs resolves the

avian family-level tree of life. *Molecular Biology and Evolution* 38: 108–127. doi: 10.1093/molbev/msaa191

Serventy, D.L. 1936. Feeding methods of *Podargus*: with remarks on the possible causes of its aberrant habits. *Emu* 36: 74–90. doi: 10.1071/MU936074

White, N.D. and M.J. Braun. 2019. Extracting phylogenetic signal from phylogenomic data: higher-level relationships of the nightbirds (Strisores). *Molecular Phylogenetics and Evolution* 141: 106611. doi: 10.1016/j.ympev.2019.106611

Decision letter (RSPB-2021-0181.R0)

23-Feb-2021

Dear Dr Navalón

I am pleased to inform you that your manuscript RSPB-2021-0181 entitled "Craniofacial development illuminates the evolution of nightbirds (Strisores)." has been accepted for publication in *Proceedings B*. Congratulations!

The referee(s) have recommended publication, but also suggest some minor revisions to your manuscript. Therefore, I invite you to respond to the referee(s)' comments and revise your manuscript. Because the schedule for publication is very tight, it is a condition of publication that you submit the revised version of your manuscript within 7 days. If you do not think you will be able to meet this date please let us know.

It is a condition of publication that data supporting your paper are made available either in the electronic supplementary material or through an appropriate repository. Please see our Data Sharing Policies <https://royalsociety.org/journals/authors/author-guidelines/#data>.

Sincerely,

Dr John Hutchinson, Editor

Associate Editor

Comments to Author:

Overall, I agree with the two reviewers that this is an excellent manuscript and is worthy of publication in Proc B. I am recommending revisions as per the criticisms of the referees. I do not think the stats require a specialist statistics referee as they are not complex stats by any means.

Reviewer #1 recommends major revisions, but for the most part these are organizational, not scientific revisions, a number of which reflect formatting for Proc B. I find all of the requested revisions to useful, but as the reviewer notes they are organizational and so I do not see these as

"major revisions" to the manuscript, but rather minor revisions. Please attend to these organizational and format criticisms and suggested edits.

Reviewer #2 recommends very minor revisions, and makes minor editorial suggestions. Please attend to all of these in revising the manuscript.

Reviewer(s)' Comments to Author:

Referee: 1

Comments to the Author(s)

I have read the paper written by Navalon and collaborators entitled: "Craniofacial development illuminates the evolution of nightbirds (Strisores)". The aim of the paper is to elucidate whether heterochronic shifts during ontogeny contribute to producing an apparently juvenilized cranial morphology in birds. More specifically, they aim to assess whether nightbird cranial morphology is related to ontogenetic changes and whether these can also explain the cranial shape differences in hummingbirds. To do so, the authors used a 3D geometric morphometric approach and comparative methods on a unique data set composed of 112 specimens belonging to 36 species as well as a subset with several ontogenetic series for 12 species.

I think at this point it is really important to highlight how difficult it is to get access to ontogenetic series and how important this kind of studies are. This study has a high potential and would be a really nice contribution to Proceedings B. However, I think the manuscript needs a little bit of work as some parts remain unclear or can be simplified. I think the introduction is the weakest part of the paper and needs to be rewritten in order to better introduce the concept and more clearly introduce the aim of the paper. Currently, the introduction reads as a set of paragraphs not really linked to each other and is sometimes repetitive. All the elements are there, but a little bit of work on it would be nice in order to better introduce the concept (nicely and beautifully synthesized on Figure S1 and S11) and also to show how different factors can impact the cranial evolution of the incredible Strisores clade. The material and methods and results can be clarified by the addition of headers synthesizing what the authors want to test using which methods. I will give more information on the changes that I would like to see in the manuscript below, but overall, I would like to see this manuscript published in Proceedings B and I mainly recommend writing and clarification revisions.

The manuscript is not in the format for Proc B, so this needs to be changed.

Introduction:

I would like to see an introduction written around figure S1 and maybe figure S11 with some modifications (and thus the figures move into the main manuscript) in order to summarize the different factors which could shape the cranial evolution. The combination of both figures is a really nice illustration of the questions addressed in this paper on whether altriciality is related to both accelerated cranial development and increased overall lability of development.

I would start the introduction with the explanation of the main concepts and on how development can impact morphological evolution. Move directly onto the questions addressed in this study and the presentation of the amazing model group. In the current introduction, the predictions concerning the impact of the different life history traits (highly altricial, altricial) as well as the other factors (such as specialized functional demands of nectarivory with high metabolism and so on) are not clearly made and are really missing. Yet, this is needed to understand the rest of the manuscript. Furthermore, because of the lack of a connection between each paragraph, I have the feeling that a new question is addressed into each paragraph. Then I would like to see the state of the art concerning this question. Here the authors can highlight that nearly no studies have looked at ontogenetic series and this is why their study is unique. I would also state why nearly no studies are looking at this. It is easy to say that other studies look at adults only, but it is also because ontogenetic series are not abundant in collections (which makes this kind of study complicated to realize).

I would end the introduction with a paragraph on what they will more specifically investigate and the predictions of the results following the literature. These predictions will help to structure the method section: for example: species with strong developmental constraints will show a lower disparity than those that are less constraint, and so one).

Line 47: references are needed

Line 43: the way that the word macroevolution is used here doesn't make any sense It should be use as 'pattern of evolution at the macroevolutionary scale'

Line 69: "hummingbird" need to be replace by hummingbird

MATERIAL AND METHODS:

I would mainly add headers summarizing what the authors want to test using which methods. For each section I would start by the aim of the analyses prior to giving the detail on the analyses. This will make the reading of the methods easier and will make it more straightforward to understand why which kind of method is performed and why.

I would move the phylogeny section into the methods section (it is currently in material).

One part needs to be clarified concerning the test of convergence / divergence. I may have missed something while reading, but it is unclear to me how the authors can test for shape convergence/ divergence while using disparity tests. For me, they can only assess difference in morphological diversity using this kind of test (see below for more details).

Line 117: change "followed" by "used"

Paragraph line 119-122: need to be rephrased in order to better link both sentences. Moreover, 'skewed' is used twice in the same sentence

Line 127: the script provided in supplementary information is not up to date in comparison to what they write in the main manuscript (the authors used the "plotTangentSpace" function in the script, which has been recently deprecated, and is now replaced by the "gm.prcomp" function). As the geomorph R package changes the name of their functions really often and the authors only used published functions in this manuscript, I don't see the point to provide the script. It may be confusing for the reader if they want to reproduce the analyses in few months, the script may be quickly unusable by anybody due to the changes of the names of the functions. Furthermore, nobody is asking to see screenshot of the button to use while they are using "PAST".

Line 164: the authors used a test of disparity in order to assess ontogenetic convergence/ divergence pattern among clades. Maybe I misunderstood something, but by doing that, I have the feeling that they can only assess whether the disparity is higher or lower in one clade in comparison to another (so just difference in shape variance and not difference in shape). To test that I would either do a test comparing the slope, or do a test of convergence (Stayton test or RRphylo test of convergence).

Line 167: function plotGMPhyloMorphoSpace is now Deprecated and defunct functions in geomorph

Line 172 "focussing" needs to be changed by "focusing"

RESULTS

Adding headers as in method section can be helpful.

DISCUSSION

Line 282 and 285: there is no fig. 4 in the manuscript.

Line 301: extra space between "(1-3)" and "."

Figure 1: the use of the letters (a., b., c., d....) in both phylogeny+phyomorphospace (node labeling) and in the overall figure is confusing. It is possible to change them in the phylogeny+phyomorphospace by capitalized letters?

Supplemental information:

Table S3, the table caption is before the table whereas it is placed after for all the other tables.

Referee: 2

Comments to the Author(s)

This is an extremely interesting study that will contribute greatly to our understanding of the evolution and development of an unusual clade of birds, with implications for the macroevolution of vertebrates in general. The figures are well rendered and the interpretations seem to be reasonable given the findings presented. Accordingly, I have only a few minor concerns.

Firstly, I appreciate the repeated acknowledgements that the internal relationships within Strisores are controversial. However, I am not aware of any recent study that has recovered the topology used in this study, whereas White and Braun (2019) and Kuhl et al. (2021) recovered topologies congruent with those of Prum et al./Chen et al., and the preferred topology of Braun and Kimball (2021) differs from those only in the position of *Steatornis*. Given that these changes in tree topology are unlikely to change the study's conclusions (as stated in the supplementary material), I would not insist on rerunning the analyses under alternative phylogenetic hypotheses, but I recommend that this is given greater emphasis in the manuscript, preferably with at least a brief mention in the main text, e.g., "Although the phylogenetic hypothesis adopted here has been disputed by recent studies, alternative topologies would not change our conclusions (Supplementary Information)."

Although the discussion understandably focuses on hummingbird origins, the findings on frogmouths are also interesting, and I think further discussion of these results (at least in supplementary material) would make a good addition to the manuscript. In particular, it is mentioned several times that frogmouths exhibit only a "partial" reversal to the ancestral ontogenetic pattern. I would be interested in seeing further elaboration on what this entails (i.e. in what ways is frogmouth ontogeny similar to/different from those of other strisoreans/non-strisorean birds?). Also worthy of discussion might be potential implications of this finding for previous speculations that frogmouths evolved from aerial hawkers (Serventy, 1936; Chen et al.) and the timing of this transition based on fossils of stem-frogmouths (*Masillapodargus* and possibly *Fluvioviridavis*).

I find it odd that caprimulgids are color-coded as semi-altricial in Fig. S10, but are (to my knowledge correctly) listed as semi-precocial in Supplementary Dataset 2B. It is also not immediately clear to me what the arrows on the leftmost side of the figure indicate. Perhaps an explanation in the caption is warranted here.

Lastly, I offer a list of grammatical, stylistic, or typographic suggestions:

Line 65: The comma after "configuration" is not needed.

Line 74: I suggest "developmental variations".

Line 241: The comma after "extent" is not needed.

Line 266: Looks like a closing parenthesis is needed after citation (40).

Line 273: Should read "leading to hummingbirds".

Line 275: Should read "may have".

Fig. S4a: I suggest changing “most youngest specimens” to “the majority of the youngest specimens” (as in S4b), as the latter sounds less confusing.

Fig. S11: I recommend “In green &are alterations [...]” and “In grey &are ecological [...]”, as this seems to read more clearly.

Supplementary Dataset 2B: Plumage details for &Eulampis holosericeus& should read “Pinkish &chick&”, not “Pinkish chicken”.

&References&

Braun, E.L. and R.T. Kimball. 2021. Data types and the phylogeny of Neoaves.

&Birds& 2: 1–22. doi: 10.3390/birds2010001

Kuhl, H., C. Frankl-Vilches, A. Bakker, G. Mayr, G. Nikolaus, S.T. Boerno, S. Klages, B.

Timmermann, and M. Gahr. 2021. An unbiased molecular approach using 3'UTRs resolves the avian family-level tree of life. &Molecular Biology and Evolution& 38: 108–127. doi: 10.1093/molbev/msaa191

Serventy, D.L. 1936. Feeding methods of &Podargus&: with remarks on the possible causes of its aberrant habits. &Emu& 36: 74–90. doi: 10.1071/MU936074

White, N.D. and M.J. Braun. 2019. Extracting phylogenetic signal from phylogenomic data: higher-level relationships of the nightbirds (Strisores). &Molecular Phylogenetics and Evolution& 141: 106611. doi: 10.1016/j.ympev.2019.106611

Author's Response to Decision Letter for (RSPB-2021-0181.R0)

See Appendix A.

Decision letter (RSPB-2021-0181.R1)

17-Mar-2021

Dear Dr Navalón

I am pleased to inform you that your manuscript entitled "Craniofacial development illuminates the evolution of nightbirds (Strisores)." has been accepted for publication in Proceedings B.

Your article has been estimated as being 9 pages long. Our Production Office will be able to confirm the exact length at proof stage.

Data Accessibility section

Open Access

Paper charges

Sincerely,

Proceedings B

Appendix A

Dr John Hutchinson, Editor

Associate Editor

Comments to Author:

Overall, I agree with the two reviewers that this is an excellent manuscript and is worthy of publication in Proc B. I am recommending revisions as per the criticisms of the referees. I do not think the stats require a specialist statistics referee as they are not complex stats by any means.

Reviewer #1 recommends major revisions, but for the most part these are organizational, not scientific revisions, a number of which reflect formatting for Proc B. I find all of the requested revisions to useful, but as the reviewer notes they are organizational and so I do not see these as "major revisions" to the manuscript, but rather minor revisions. Please attend to these organizational and format criticisms and suggested edits.

Reviewer #2 recommends very minor revisions, and makes minor editorial suggestions. Please attend to all of these in revising the manuscript.

Reviewer(s)' Comments to Author:

Referee: 1

Comments to the Author(s)

I have read the paper written by Navalon and collaborators entitled: "Craniofacial development illuminates the evolution of nightbirds (Strisores)". The aim of the paper is to elucidate whether heterochronic shifts during ontogeny contribute to producing an apparently juvenilized cranial morphology in birds. More specifically, they aim to assess whether nightbird cranial morphology is related to ontogenetic changes and whether these can also explain the cranial shape differences in hummingbirds. To do so, the authors used a 3D geometric morphometric approach and comparative methods on a unique data set composed of 112 specimens belonging to 36 species as well as a subset with several ontogenetic series for 12 species.

I think at this point it is really important to highlight how difficult it is to get access to ontogenetic series and how important this kind of studies are. This study has a high potential and would be a really nice contribution to Proceedings B. However, I think the manuscript needs a little bit of work as some parts remain unclear or can be simplified. I think the introduction is the weakest part of the paper and needs to be rewritten in order to better introduce the concept and more clearly introduce the aim of the paper. Currently, the introduction reads as a set of paragraphs not really linked to

each other and is sometimes repetitive. All the elements are there, but a little bit of work on it would be nice in order to better introduce the concept (nicely and beautifully synthesized on Figure S1 and S11) and also to show how different factors can impact the cranial evolution of the incredible Strisores clade. The material and methods and results can be clarified by the addition of headers synthesizing what the authors want to test using which methods. I will give more information on the changes that I would like to see in the manuscript below, but overall, I would like to see this manuscript published in Proceedings B and I mainly recommend writing and clarification revisions.

The manuscript is not in the format for Proc B, so this needs to be changed.

Amended now.

Introduction:

I would like to see an introduction written around figure S1 and maybe figure S11 with some modifications (and thus the figures move into the main manuscript) in order to summarize the different factors which could shape the cranial evolution. The combination of both figures is a really nice illustration of the questions addressed in this paper on whether altriciality is related to both accelerated cranial development and increased overall lability of development.

I would start the introduction with the explanation of the main concepts and on how development can impact morphological evolution. Move directly onto the questions addressed in this study and the presentation of the amazing model group. In the current introduction, the predictions concerning the impact of the different life history traits (highly altricial, altricial) as well as the other factors (such as specialized functional demands of nectarivory with high metabolism and so one) are not clearly made and are really missing. Yet, this is needed to understand the rest of the manuscript. Furthermore, because of the lack of a connection between each paragraph, I have the feeling that a new question is addressed into each paragraph.

Then I would like to see the state of the art concerning this question. Here the authors can highlight that nearly no studies have looked at ontogenetic series and this is why their study is unique. I would also state why nearly no studies are looking at this. It is easy to say that other studies look at adults only, but it is also because ontogenetic series are not abundant in collections (which makes this kind of study complicated to realize).

I would end the introduction with a paragraph on what they will more specifically investigate and the predictions of the results following the literature. These predictions will help to structure the method section: for example: species with strong developmental constraints will show a lower disparity than those that are less constraint, and so one).

We are very grateful for this constructive and very useful break-down on how to reframe our Introduction. We followed the suggested structure in full and we considered the new text is substantially improved. However, for the sake of brevity we did not incorporate specific predictions regarding our working hypotheses as this can be easily distilled from the new Figure 1, which visually displays the aforementioned predictions, and includes visual cues in the form of colours. For

instance, the idealised ontogenetic trajectory showing the expectation of one kind of paedomorphism (progenesis) is in the same orange/ochre colour as the one used for the broad skull clades hypothesised to have experienced this mechanism of ontogenetic evolution throughout the manuscript. The same happens with the green colour used for hummingbirds. Additionally, we added two silhouettes and two question marks next to them to further stress this whilst maintained a clean layout (explained in the captions). Similarly, we did not delve too much into explicit predictions of variation in life history or ecological traits as these relationships can be seen in the new Fig. 1 in the context of hummingbird evolution and we only succinctly refer to them in the text.

Line 47: references are needed

Line 43: the way that the word macroevolution is used here doesn't make any sense It should be use as 'pattern of evolution at the macroevolutionary scale'

We removed this part of the Introduction concerning patterns in other amniote groups to refocus the Introduction in the group and specific aims of the article as suggested above by Reviewer 1.

Line 69: "hummgbird" need to be replace by hummingbird

Done.

MATERIAL AND METHODS:

I would mainly add headers summarizing what the authors want to test using which methods. For each section I would start by the aim of the analyses prior to giving the detail on the analyses. This will make the reading of the methods easier and will make it more straightforward to understand why which kind of method is performed and why.

We have made significant changes in M&M section following the reviewer suggestion of explaining the aim of each analysis prior to the technical details. Hopefully, everything is clearer.

I would move the phylogeny section into the methods section (it is currently in material).

Done.

One part needs to be clarified concerning the test of convergence / divergence. I may have missed something while reading, but it is unclear to me how the authors can test for shape convergence/ divergence while using disparity tests. For me, they can only assess difference in morphological diversity using this kind of test (see below for more details).

We compared disparity between earliest and adult individuals. For a set of ontogenetic trajectories, if the earliest individuals exhibit less disparity than the adults one can say the ontogenies are diverging; if the earliest stages are more disparate than the adults one can say the ontogenies are converging. However, we agree it is a tad ambiguous as it does not describe convergence in a more

strict sense. Also, we compared among adult groups so we fully acknowledge this is not the way of portraying this analysis and rephrase all relevant text.

Line 117: change “followed” by “used”

Done.

Paragraph line 119-122: need to be rephrased in order to better link both sentences. Moreover, ‘skewed’ is used twice in the same sentence

We added an additional phrase and some bits of text to hopefully clarify this part of the methods.

Line 127: the script provided in supplementary information is not up to date in comparison to what they write in the main manuscript (the authors used the “plotTangentSpace” function in the script, which has been recently deprecated, and is now replaced by the “gm.prcomp” function). As the geomorph R package changes the name of their functions really often and the authors only used published functions in this manuscript, I don’t see the point to provide the script. It may be confusing for the reader if they want to reproduce the analyses in few months, the script may be quickly unusable by anybody due to the changes of the names of the functions. Furthermore, nobody is asking to see screenshot of the button to use while they are using “PAST”.

We absolutely agree with the reviewer in this point. Considering we did not write any custom code for this research we will not provide the R code unless indicated otherwise by the editorial board.

Line 164: the authors used a test of disparity in order to assess ontogenetic convergence/ divergence pattern among clades. Maybe I misunderstood something, but by doing that, I have the feeling that they can only assess whether the disparity is higher or lower in one clade in comparison to another (so just difference in shape variance and not difference in shape). To test that I would either do a test comparing the slope, or do a test of convergence (Stayton test or RRphylo test of convergence).

See our comments above.

Line 167: function plotGMPhyloMorphoSpace is now Deprecated and defunct functions in geomorph.

We change this to the new function gm.prcomp which is exactly analytically equivalent to the previous function.

Line 172 “focussing” needs to be changed by “focusing”

Done.

RESULTS

Adding headers as in method section can be helpful.

DISCUSSION

Line 282 and 285: there is no fig. 4 in the manuscript.

Changed to new Fig. 1.

Line 301: extra space between “(1-3)” and “.”

Thanks! Amended.

Figure 1: the use of the letters (a., b., c., d.....) in both phylogeny+phyomorphospace (node labeling) and in the overall figure is confusing. It is possible to change them in the phylogeny+phyomorphospace by capitalized letters?

We changed clade letters to capital letters.

Supplemental information:

Table S3, the table caption is before the table whereas it is placed after for all the other tables.

Captions before some large tables are there because we lack space for it below whilst maintaining a sufficiently large size for the table to be completely legible. However, this is indicated in the caption.

Referee: 2

Comments to the Author(s)

This is an extremely interesting study that will contribute greatly to our understanding of the evolution and development of an unusual clade of birds, with implications for the macroevolution of vertebrates in general. The figures are well rendered and the interpretations seem to be reasonable given the findings presented. Accordingly, I have only a few minor concerns.

Firstly, I appreciate the repeated acknowledgements that the internal relationships within Strisores are controversial. However, I am not aware of any recent study that has recovered the topology used in this study, whereas White and Braun (2019) and Kuhl et al. (2021) recovered topologies congruent with those of Prum et al./Chen et al., and the preferred topology of Braun and Kimball (2021) differs from those only in the position of Steatornis. Given that these changes in tree topology are unlikely to change the study's conclusions (as stated in the supplementary material), I would not insist on rerunning the analyses under alternative phylogenetic hypotheses, but I recommend that this is given greater emphasis in the manuscript, preferably with at least a brief mention in the main text, e.g., "Although the phylogenetic hypothesis adopted here has been disputed by recent studies, alternative topologies would not change our conclusions (Supplementary Information)."

Our supertree uses the backbone from Hackett et al., 2008 available at birdtree.org. This is enforced by the way the supertree was constructed (see Methods). However, we agree with the reviewer and we did the suggested changes in the main text.

Although the discussion understandably focuses on hummingbird origins, the findings on frogmouths are also interesting, and I think further discussion of these results (at least in supplementary material) would make a good addition to the manuscript. In particular, it is mentioned several times that frogmouths exhibit only a “partial” reversal to the ancestral ontogenetic pattern. I would be interested in seeing further elaboration on what this entails (i.e. in what ways is frogmouth ontogeny similar to/different from those of other strisoreans/non-strisorean birds?). Also worthy of discussion might be potential implications of this finding for previous speculations that frogmouths evolved from aerial hawkers (Serventy, 1936; Chen et al.) and the timing of this transition based on fossils of stem-frogmouths (Masillapodargus and possibly *Fluvioviridavis*).

We find this suggestion very interesting and although we cannot include our reflections on this topic in the main text for the sake of narrative coherence and brevity, we did include a paragraph in the extended results.

I find it odd that caprimulgids are color-coded as semi-altricial in Fig. S10, but are (to my knowledge correctly) listed as semi-precocial in Supplementary Dataset 2B. It is also not immediately clear to me what the arrows on the leftmost side of the figure indicate. Perhaps an explanation in the caption is warranted here.

Although it is true that HBW text suggest some caprimulgids are semi-precocial, our data (e.g., Botelho et al., 2015; Botelho and Faunes, 2015; which are mostly derived from Starck classic papers on the altricial-precocial spectrum), footage and real observations suggests caprimulgids are indeed semi-altricial as they are locomotory underdeveloped upon hatching (see Table). We recognised this can be conflated with the tendency for immobility in cryptic floor nesting species such as most caprimulgids. For the time being we changed our observation to semi-altricial in SM. Dataset B. If the reviewer has got other additional information we are happy to retract our views and fix this in a different direction. However, the mixed life history categories (semi-altricial and semi-precocial) could easily be recategorized as an inclusive mixed category as the boundaries among both life histories are likely to be blurry. Arrows indicate species for which ontogenetic data is available in this study, we added this to the caption.

Lastly, I offer a list of grammatical, stylistic, or typographic suggestions:

Line 65: The comma after “configuration” is not needed.

Line 74: I suggest “developmental variations”.

Line 241: The comma after “extent)” is not needed.

Line 266: Looks like a closing parenthesis is needed after citation (40).

Line 273: Should read “leading to hummingbirds”.

Line 275: Should read “may have” .

Fig. S4a: I suggest changing “most youngest specimens” to “the majority of the youngest specimens” (as in S4b), as the latter sounds less confusing.

Fig. S11: I recommend “In green are alterations [...]” and “In grey are ecological [...]”, as this seems to read more clearly.

Supplementary Dataset 2B: Plumage details for *Eulampis holosericeus* should read “Pinkish chick”, not “Pinkish chicken”.

Thanks for this! We fixed all.

References

Braun, E.L. and R.T. Kimball. 2021. Data types and the phylogeny of Neoaves. *Birds* 2: 1–22. doi: 10.3390/birds2010001

Kuhl, H., C. Frankl-Vilches, A. Bakker, G. Mayr, G. Nikolaus, S.T. Boerno, S. Klages, B. Timmermann, and M. Gahr. 2021. An unbiased molecular approach using 3'UTRs resolves the avian family-level tree of life. *Molecular Biology and Evolution* 38: 108–127. doi: 10.1093/molbev/msaa191

Serventy, D.L. 1936. Feeding methods of *Podargus*: with remarks on the possible causes of its aberrant habits. *Emu* 36: 74–90. doi: 10.1071/MU936074

White, N.D. and M.J. Braun. 2019. Extracting phylogenetic signal from phylogenomic data: higher-level relationships of the nightbirds (Strisores). *Molecular Phylogenetics and Evolution* 141: 106611. doi: 10.1016/j.ympev.2019.106611